# Predisposition to Myocardial Infarction Influenced by Interleukin 13 Gene Polymorphisms: A Case-Control Study

**DOI:** 10.3390/genes13081478

**Published:** 2022-08-19

**Authors:** Seyyed Fatemeh Hosseini, Khalil Khashei Varnamkhasti, Raziyeh Naeimi, Leila Naeimi, Sirous Naeimi

**Affiliations:** 1Department of Genetics, College of Science, Kazerun Branch, Islamic Azad University, Kazerun 73, Iran; 2Department of Cardiology, School of Medicine, Shiraz University of Medical Sciences, Shiraz 73, Iran

**Keywords:** myocardial infarction, *Interleukin-13* gene, polymorphism, *IL-13* −1512A/C, *IL-13* +2044G/A

## Abstract

Background: Additional inflammatory responses and subsequent damage—arising from enhance transcriptional activity or forming the more active protein due to existence of polymorphic sites in the pro-inflammatory cytokines gene loci—give rise to myocardial infarction susceptibility. Objectives: The aim of our study was to explore whether two *interleukin-13* gene polymorphisms (−1512A/C and +2044G/A) could serve as underpins genetic susceptibility of myocardial infarction. Methods: The Iranian population that belong to the Parsis ethnic group was involved in the present study. A total 250 patients with definite myocardial infarction—meeting hypertension, hypercholesterolemia, hyperglycemia, and coronary artery disease requirements—were recruited from the Shiraz urban hospitals. 250 age- and sex-matched healthy individuals without a history of cardiovascular disease and heart disease related risk factors constituted the control group. PCR-restriction fragment length polymorphism technique applied to genotyping at −1512A/C and +2044G/A loci. Hardy–Weinberg equilibrium test was performed (combined cases and controls). The differences of the genotype frequencies in cases and controls were analyzed using a chi-square test. Logistic regression analysis was performed to assess the association between the genotypes and most important risk factors for myocardial infarction. All statistical analyses were performed in SPSS Version 22.0. *p*-values below 0.05 were hailed as statistically significant. Results: Deviation from Hardy–Weinberg equilibrium was not significant in the −1512A/C locus. Statistically significant difference between our study groups was found in genotype frequency of the −1512A/C. This variant was found in associated with myocardial infarction risk factors. The +2044G/A polymorphism was not in Hardy–Weinberg equilibrium and no significant difference observed in the distribution of +2044G/A genotype frequency among cases and controls. However, further analysis revealed that this genotype associated with an increased susceptibility to myocardial infarction risk factors. Conclusions: The presence of *interleukin-13* −1512A/C and +2044G/A gene polymorphisms underpin myocardial infarction predisposition in the ethnic Parsis of the Iranian population.

## 1. Introduction

Myocardial infarction (MI), what is colloquially known as ‘heart attack’, still carries 5–30% of fatality rate from out-of-hospital cardiac arrest, which caused by abrupt occlusion of the coronary arteries; the major responsible for the reduced supply of oxygen- and nutrient-rich blood to the myocardium (heart muscle) [1,2]. Coronary arterial clogging can potentially occur as a result of atherosclerotic plaque, which develops slowly from local lesion build up in artery walls. Atherosclerotic plaque is composed of a fibrous or collagenous cap (consisting of collagen as a strong material with barrier function between the bloodstream and the lipid core) and a lipid (fatty) core, made up of thrombogenic components and other substances [3,4]. Physical disruption of the fibrous cap allows direct contact between blood coagulation factors and thrombogenic components in the plaque’s lipid core, thereby triggering a chain reaction resulting in sudden obstruction of the coronary artery and myocardial infarction [4]. Much evidence implicates an inflammation related rupture of the fibrous cap [5,6]. Inflammation thins the fibrous cap and plaque prone to rupture. Inflammation also interferes with the collagen synthesis that is necessary for damaged fibrous cap repair process. When the fibrous cap becomes extremely thin, it ruptures, and coronary arteries become blocked [4]. Cytokines—as key mediators of inflammatory pathways—can lead to the destabilization of atherosclerotic plaques. Multiple lines of evidence indicate that increased inflammatory cytokine levels and inflammatory cytokine activity responsible for cardiac events may occur by polymorphisms in genes encoding inflammatory cytokines. Interleukin-13 is a Th-2-derived pro-inflammatory cytokine which its genomic locus harbors numerous single nucleotide polymorphisms (5q23- q31 segment) [7,8]. The latest analysis showed that *IL-13* −1512A/C (rs1881457) polymorphism—which resides in the distal *IL-13* promoter—upregulates HS4-dependent IL-13 transcription by creating a binding site for the transcription factor Oct-1 [9]. Furthermore, +2044G/A—a polymorphism in *IL-13* exons 4—leads to forming a novel *IL-13* variant with more STAT6 phosphorylation-inducing and monocyte CD23 expression activity than normal *IL-13* [10]. This evidence prompted us to design the present case-control study to find out the impact of genetic polymorphisms in *IL-13* gene (−1512A/C and +2044G/A) which might influence myocardial infraction predisposition.

## 2. Methods

### 2.1. Study Population and Sample Collection

This hospital-based, case-control study was performed at Shiraz urban hospitals, Iran (Namazi, Shahid Faghihi, Shahid Beheshti and, Kowsar Heart Hospital) from January 2019 to April 2021. A total of 250 patients with confirmed myocardial infarction who were diagnosed with hypertension, hypercholesterolemia, hyperglycemia, and/or coronary artery disease were compared with 250 age- and sex-matched healthy individuals with no history of cardiovascular and heart disease related risk factors (even in first-degree relatives) from Iranian population (Parsis ethnic group) (Table 1). Exclusion criteria were defined as congenital heart defects, known as structural heart disease and electrolyte disturbances. This study protocol was approved by Islamic Azad University Kazerun Branch Ethics Committee (IR.IAU.KAU.REC.1398.045) and written informed consent was provided by all the participants before entering the study groups.

### 2.2. Interleukin 13 Genotyping

Salting out procedure was employed for extracting genomic DNA from collected peripheral blood leukocytes (5 mL) on EDTA-coated tubes using a commercial genomic DNA Isolation Kit (GeNet Bio, Daejeon, Korea). Isolated genomic DNA was stored at −80 °C until genotyping was performed. Genotyping of −1512A/C and +2044G/A polymorphisms was performed by PCR-Restriction Fragment Length Polymorphism (PCR-RFLP) method, in a 15.2 μL final reaction mixture volume, utilizing specific primer pairs designed using Oligo7 software (version 7.54, Molecular Biology Insights Inc., Cascade, CO, USA).

Restriction digestion of two polymorphic sites, Bsh1236I (−1512A/C) and NlaIV (+2044G/A), on the selected, amplified PCR products were carried out for 16 h at 37 °C. Generated restriction banding patterns detected under UV light on 2% agarose gel electrophoresis followed staining by ethidium bromide.

### 2.3. Statistical Analysis

SPSS 22.0 software applied for statistical analysis (SPSS Inc., Chicago, IL, USA). Hardy–Weinberg equilibrium was used to determine consistency of genotype distribution. The differences in the genotype frequencies of each polymorphism between the case and control groups were tested using a chi-square (*X*^2^) test with one degree of freedom. The association between the genotypes and myocardial infarction’s most important risk factors were analyzed by logistic regression analysis. A *p*-value of <0.05 was hailed as statistically significant.

## 3. Results

We first evaluated the Hardy–Weinberg equilibrium by computing expected genotype values versus observed genotype values for both polymorphic loci to check whether the population was in Hardy–Weinberg equilibrium. The results showed that the deviation from Hardy–Weinberg equilibrium in the −1512A/C locus was not significant, therefore equilibrium was maintained for in question population at polymorphic −1512A/C site. The distribution of genotypes in +2044G/A locus suggest significant deviation and departure from Hardy–Weinberg equilibrium.

We genotyped the −1512A/C and +2044G/A polymorphisms in 250 patients present with myocardial infarction and 250 age- and sex-similar healthy control subjects. As displayed in Table 2, significant difference was found between patients with myocardial infraction and healthy controls in −1512A/C polymorphism genotype frequency (*p* = 0.031). However, the +2044G/A polymorphism genotype frequency did not reach statistical significance (*p* = 0.349).

We also evaluated the influence of the both in question polymorphisms on hypertension and other myocardial infraction risk factors such as hypercholesterolemia, hyperglycemia, and coronary artery disease. The following table (Table 3) shows a significant association between the *IL-13* −1512A/C and +2044G/A polymorphisms and increased susceptibility to risk factors which develop myocardial infraction. Moreover, our findings propose a significant association between 1512A/C and +2044G/A polymorphisms and genetic ancestry.

## 4. Discussions

From the existing fact that most diseases are influenced by DNA sequence variations, a study of single nucleotide base substitution can assess the possible role of polymorphisms in disease susceptibility, as well as disease progression [11,12]. This article points out single nucleotide base substitutions in *IL**-13* gene’s promoter (−1512A/C) and coding regions (+2044G/A) which were found as genetic factors associated with myocardial infraction predisposition and progression. The −1512A/C locus was found to be in Hardy–Weinberg equilibrium and its genotype frequencies were significantly different between MI patients and controls. All variant genotypes—AA, AC, and CC—were significantly associated with myocardial infarction and its major risk factors. Possible correlation between *IL-13* promoter rs 1,881,457 genotype and heart (cardiovascular) disease has been followed in only one previous study conducted by Zha et al. In this empirically study, two-fold increased risk of coronary artery disease mediated by rs1881457 variant reported in men [13]. The rs1881457 variant effects on hypertension also have been observed in the Korean population [14]. Another candidate *IL**-13* gene polymorphism, +2044G/A, deviated from Hardy–Weinberg equilibrium with no observing significant differences in its genotype frequencies (GG, GA, and AA) among MI patients and controls. Although results suggested departure of +2044G/A from HWE but risk factor-analysis revealed that myocardial infarction development influenced by all three genotypes of +2044G/A polymorphism. Various studies have been confirmed the role of +2044G/A polymorphism in immune-mediated disorders. For example; Liao et al. study suggested that rs20541 is associated with the risk of chronic obstructive pulmonary disease [15]. The higher risk of asthma associated with rs20541 genotype has been reported in a study conducted by Halwani et al. in the Saudi Arabian population [16]. Contribution of rs20541 variant to the risk of coronary artery disease has also been noted by Zha et al. [13]. IL-13 protein (coded by *IL-13* gene) is a key inducer of many pathological processes and this study suggests a significant value of its polymorphisms in diagnosis and risk stratification in the myocardial infraction individual patient. Genetic testing is expected to afford the opportunity for assessment of an individual’s risk profile, not only in members of high-risk families (according to ESC guidelines) but also in the general population, as part of routine health care. Moreover, the population screening method based on detecting single nucleotide base substitutions is a cost-effective method for use in clinical practice.

## 5. Limitations

There are some noteworthy limitations to this study. First, our study was validated in a relatively small sample-size; therefore, larger sample sizes are needed for further validation. Second, due to the limitations of sample access from other ethnic groups, study subjects were from a single ethnic group. Third, although we applied a rigorous design in selecting study subjects in order to mitigate all possible biases in the study, inherent selection bias cannot be completely excluded.

## 6. Conclusions

As such, the known single nucleotide base substitutions which play an important role in the pathogenesis of myocardial infarction could be used as new genetic diagnostic markers in susceptibility (or predisposition) testing.

## Figures and Tables

**Table 1 genes-13-01478-t001:** Demographic characteristics of MI patients and controls.

Variables	MI Patients	Controls	*p*-Value
	N = (250)	N = (250)	
Age, years	57.54 ± 8.51	57.82 ± 8.07	0.239
range	33–70	35–70	-
Sex (Men)	129	129	0.142
Hypertension n (%)	148 (59.2)	-	-
Hypercholesterolemia n (%)	192 (76.8)	-	-
Hyperglycemia n (%)	167 (66.8)	-	-
Coronary artery disease n (%)	148 (42.8)	-	-

**Table 2 genes-13-01478-t002:** Genotype frequency distributions of −1512A/C and +2044G/A polymorphisms in MI patients and controls.

*IL-13*	MI Patients n (%)	Controls n (%)	*p*-Value
rs1881457 (−1512A/C)			
AA	147 (58.8)	120 (48)	0.031
AC	92 (36.8)	110 (44)	
CC	11 (4.4)	20 (8)	
rs20541 (+2044G/A)			
GG	136 (54.4)	121 (48.4)	0.349
GA	90 (36)	98 (39.2)	
AA	24(9.6)	31 (12.4)	

**Table 3 genes-13-01478-t003:** Associations of −1512A/C and +2044G/A polymorphisms and the development of myocardial infraction risk factors.

** *IL-13* **	**Genotypes** **n (%)**		
	**AA**	**AC**	**CC**
rs1881457 (−1512A/C)			
Hypercholesterolemia			
>200 mg	122 (48.8)	63 (25.2)	7 (2.8)
<200 mg	25 (10)	29 (11.6)	24 (1.6)
Sig. (Exp. (B))	0.041 (2.779)	0.010 (2.246)	0.000 (0.205)
Hyperglycemia			
>140 mg	112 (44.8)	47 (18.8)	8 (3.2)
<140 mg	35 (10)	45 (11.6)	3 (1.6)
Sig. (Exp. (B))	0.016 (3.036)	0.010 (1.246)	0.043 (0.205)
Hypertension			
Yes	95 (38)	51 (20.4)	2 (8)
No	52 (20.8)	41 (16.4)	9 (4.5)
Sig. (Exp. (B))	0.008 (8.221)	0.015 (1.465)	0.000 (0.547)
Coronary artery disease			
Yes	115 (46)	26 (10.4)	2 (8)
No	32 (12.8)	66 (26.4)	9 (3.6)
Sig. (Exp. (B))	0.001 (0.062)	0.000 (0.010)	0.000 (0.593)
Genetic ancestry			
Yes	144 (57.6)	78 (85.4)	10 (4)
No	3 (20.8)	14 (16.4)	1 (4.5)
Sig. (Exp. (B))	0.001 (8.671)	0.012 (4.850)	0.018 (0.21)
** *IL-13* **	**Genotypes** **n (%)**		
	**GG**	**GA**	**AA**
rs20541(+2044G/A)			
Hypercholesterolemia			
>200 mg	93 (37.2)	80 (32)	19 (7.6)
<200 mg	43 (48.4)	10 (39.2)	5 (12.4)
Sig. (Exp. (B))	0.029 (1.757)	0.021(0475)	0.008 (0.263)
Hyperglycemia			
>140 mg	79 (31.6)	70(28)	18 (7.2)
<140 mg	57 (22.8)	20(8)	6 (2.4)
Sig. (Exp. (B))	0.012 (2.165)	0.077 (0.857)	0.020 (0.333)
Hypertension			
Yes	71 (28.4)	62 (24.8)	15 (6)
No	65 (22.8)	28 (11.2)	9 (3.6)
Sig. (Exp. (B))	0.035 (1.526)	0.055(0.753)	0.022 (0.600)
Coronary artery disease			
Yes	75 (30)	15(6)	17 (6.8)
No	61 (24.4)	75(30)	7 (2.8)
Sig. (Exp. (B))	0.013 (1.975)	0.000 (0.857)	0.048 (0.412)
Genetic ancestry			
Yes	128 (51.2)	66 (26.4)	0 (0)
No	8 (3.2)	24 (9.6)	24 (9.6)
Sig. (Exp. (B))	0.019 (2.685)	0.006 (0.057)	0.009 (0.5)

Sig.: significant; Exp. (B): exponentiation of the B coefficient.

## Data Availability

The data that support the findings of this study are available from the corresponding author upon reasonable request.

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
