# Peer review of "Predisposition to Myocardial Infarction Influenced by Interleukin 13 Gene Polymorphisms: A Case-Control Study"

_genes, 2022, doi:10.3390/genes13081478_

Round 1

Reviewer 1 Report

In this manuscript, the authors conduct the genetic association between two interleukin-13 genetic variants and myocardial infarction (MI) based on a hospital-based case control study. The authors included 250 MI patients and 250 age- and sex-matched control subjects for the analysis. The chi-squared test revealed that the genotype frequency of rs1881457 (-1512A/C) was statistically different between MI cases and controls. My comments are below.

·       In the abstract Objective section, the objective is unclear. The sentence for “Objective” indicates the hypothesis not the objective of this study.

·       In the abstract Methods section, the characteristics of the study population is unclear.

·       In the abstract Methods section, chi-square test is not appropriate for the genetic association study. The potential confounders should be considered in the analysis, particularly genetic ancestry. How did the authors control the population stratification, the most common confounding in the genetic association study? What was the ethnicity of the study population? Did the authors consider multiple logistic regression analysis adjusting for potential confounders?

·       In the abstract Results section, the authors mentioned that the -1512A/C polymorphism has reached Hardy-Weinberg equilibrium (HWE), which is confusing. Does the author mean that HWE test for the polymorphism was not statistically significant? More importantly, as HWE test is a procedure of quality control of genotyped variants, it is not necessary to mention in the abstract.

·       Throughout the manuscript, the grammatical errors need to be corrected.

·       In Table 1, for Sex (Men-Women) row, number for one sex is sufficient as the Table provides the total number of subjects by case and control. I would recommend presenting one number for one sex.  

·       In Table 1, P-value is confusing. Sex was matched but why p-value is significant (P<0.001)? For the other rows (hypertension, hypercholesterolemia, hyperglycemia and coronary artery disease), it does not make sense to present the p-value. What does the p-values <0.001 for those variables indicate? What was the comparison group?

·       Table II and Figure 1 are not necessary. The details for PCR procedure and gel electrophoresis figure are not necessary in this manuscript. They are quite standardized and are not informative in terms of the objective in this study.

·       In Methods, Statistical analysis section, the authors should expand the details of the statistical analysis. Hardy-Weinberg equilibrium (HWE) test should not be the main analysis for this study. HWE check is part of the quality control of variants for the genetic association analysis. The authors should have considered the genetic ancestry as the potential confounders in the statistical analysis.

·       The Table III is not necessary for this article. The HWE test is often performed in control only group with more stringent statistical significance. The authors did not mention the details of the HWE test in the Methods section. Although the authors observed the deviation of the HWE for rs20541, the authors moved forward the further analysis with this variant. The authors did not address the rational for conducting the chi-squared test for this variant deviated from HWE test.

·       Overall, the statistical analysis was not appropriate for the research question. The authors did not consider the population stratification, the major confounder in the genetic association study. The multiple logistic regression adjusting for genetic ancestry and other covariates is highly recommended for the analysis. The principal components analysis by smartpca software for the genetic ancestry is highly recommended.

·       This is the hospital-based case-control study, which is prone to selection bias. I would recommend the authors’ thoughts  about the potential selection bias in this study in the Discussion section. 

Author Response

  1. In the abstract Objective section, the objective is unclear. The sentence for “Objective” indicates the hypothesis not the objective of this study.

The aim of our study was to explore whether two interleukin-13 gene polymorphisms (−1512A/C and +2044G/A) could serve as underpins genetic susceptibility of myocardial infarction.

  1. In the abstract Methods section, the characteristics of the study population is unclear.

The Iranian population belonged to Parsis ethnic group was involved in the present study. A total 250 patients with definite myocardial infarction who meeting hypertension, hypercholesterolemia, hyperglycemia and coronary artery disease recruited from the Shiraz urban hospitals. 250 age- and sex-matched healthy individuals without a history of cardiovascular disease and heart disease related risk factors constituted the control group.

  1. In the abstract Methods section, chi-square test is not appropriate for the genetic association study. The potential confounders should be considered in the analysis, particularly genetic ancestry. How did the authors control the population stratification, the most common confounding in the genetic association study? What was the ethnicity of the study population? Did the authors consider multiple logistic regression analysis adjusting for potential confounders?

The differences of the genotype frequencies in cases and controls were analyzed using a chi-square test. Logistic regression analysis was performed to assess the association between the genotypes and myocardial infarction most important risk factors.

Due to the limitation sample access from other ethnic groups, we did not include genotype differences between different ethnic groups. The Iranian population belonged to Parsis ethnic group were involved in our study.

  1. In the abstract Results section, the authors mentioned that the -1512A/C polymorphism has reached Hardy-Weinberg equilibrium (HWE), which is confusing. Does the author mean that HWE test for the polymorphism was not statistically significant? More importantly, as HWE test is a procedure of quality control of genotyped variants, it is not necessary to mention in the abstract.

Deviation from Hardy-Weinberg equilibrium was not significant in the -1512A/C locus.

  1. In Table 1, for Sex (Men-Women) row, number for one sex is sufficient as the Table provides the total number of subjects by case and control. I would recommend presenting one number for one sex.  

Revised.

  1. In Table 1, P-value is confusing. Sex was matched but why p-value is significant (P<0.001)? For the other rows (hypertension, hypercholesterolemia, hyperglycemia and coronary artery disease), it does not make sense to present the p-value. What does the p-values <0.001 for those variables indicate? What was the comparison group?

Revised.

  1. Table II and Figure 1 are not necessary. The details for PCR procedure and gel electrophoresis figure are not necessary in this manuscript. They are quite standardized and are not informative in terms of the objective in this study.

Revised.

  1. In Methods, Statistical analysis section, the authors should expand the details of the statistical analysis. Hardy-Weinberg equilibrium (HWE) test should not be the main analysis for this study. HWE check is part of the quality control of variants for the genetic association analysis. The authors should have considered the genetic ancestry as the potential confounders in the statistical analysis.

SPSS 22.0 software applied for statistical analysis (SPSS Inc., Chicago, IL, USA). Hardy-Weinberg equilibrium was used to determine consistency of genotype distribution. The differences in the genotype frequencies of each polymorphism between the case and control groups were tested using a chi-square (X2) test with one degree of freedom. The association between the genotypes and myocardial infarction most important risk factors were analyzed by logistic regression analysis. P-value < 0.05 was hailed as statistically significant.

  1. The Table III is not necessary for this article. The HWE test is often performed in control only group with more stringent statistical significance.

Revised

  1. This is the hospital-based case-control study, which is prone to selection bias. I would recommend the authors’ thoughts  about the potential selection bias in this study in the Discussion section. 

There are some noteworthy limitations to this study. First, our study validated in relatively small sample-sized, therefore, more large sample–sized are needed to further validation. Second, due to the limitation sample access from other ethnic groups, study subjects were from single ethnic group. Third, although we applied a rigorous design in selecting study subjects to mitigating all possible biases in the study, but inherent selection bias cannot be completely excluded.

Reviewer 2 Report

The authors submitted a researhc article in which they evaluated interleukin-13 gene polymorphisms (−1512A/C and +2044G/A) in connection with genetic susceptibility of myocardial infarction They included 250 patients with acute MI and and 250 age- and sex-matched control subjects. The issue of the study was that presence of interleukin-13 −1512A/C and +2044G/A gene polymorphisms underpinned myocardial infarction predisposition. The strength of the study is a hot point of the hypothesis, the weaknes are a lack of GWAS technology support and cohort design. Although the findings appear to be interesting, I would like to put forward several issues to comments.

1. Study design: The authors included acute MI patients with CV risk factors and other individuals without CV risk factors. It seems to be logical to construct three subgroups of the individuals, for instance: acute MI, people with CV risk factros without MI, and healthy volunteers.

2. It remained unclear underlying severity of atherosclerosis of the acute MI patients. Please, check and add more information.

3.Statistics. The authors used descriptive statistics, but the aim of the study is considered to be reached by regression and factorial analysis. Please, check it cnad give extensive explanation.

4. Deiscussion: Authors should report their comment regarding the fact that ESC recommends to avoid genetic testinf in CVD apart from limiting cohorts of patients.

5. Cost of the implementation of the findings into practice. Please, give your opinion about it.

Author Response

  1. Study design: The authors included acute MI patients with CV risk factors and other individuals without CV risk factors. It seems to be logical to construct three subgroups of the individuals, for instance: acute MI, people with CV risk factros without MI, and healthy volunteers.

The study includes two groups;

Group1. Patients with confirmed myocardial infarction who meeting hypertension, hypercholesterolemia, hyperglycemia and coronary artery disease

Group 2. Healthy individuals with no history of cardiovascular disease and heart disease related risk factors (even in first-degree relatives)

  1. It remained unclear underlying severity of atherosclerosis of the acute MI patients. Please, check and add more information.

Coronary artery disease includes atherosclerosis.

3.Statistics. The authors used descriptive statistics, but the aim of the study is considered to be reached by regression and factorial analysis. Please, check it cnad give extensive explanation.

SPSS 22.0 software applied for statistical analysis (SPSS Inc., Chicago, IL, USA). Hardy-Weinberg equilibrium was used to determine consistency of genotype distribution. The differences in the genotype frequencies of each polymorphism between the case and control groups were tested using a chi-square (X2) test with one degree of freedom. The association between the genotypes and myocardial infarction most important risk factors were analyzed by logistic regression analysis. P-value < 0.05 was hailed as statistically significant.

  1. Deiscussion: Authors should report their comment regarding the fact that ESC recommends to avoid genetic testinf in CVD apart from limiting cohorts of patients.

Genetic testing is expected to afford the opportunity for assessment of an individual’s risk profile, not only in members of high-risk families (According to ESC guidelines) but also in the general population, as part of routine health care.

  1. Cost of the implementation of the findings into practice. Please, give your opinion about it.

Population screening method based on detecting single nucleotide base substitutions is a cost-effective method for used in clinical practice.  

Round 2

Reviewer 1 Report

1. The authors still did not consider the genetic ancestry in the model. I highly recommend conducting the principal component analysis (the researchers often use ‘smartpca’ software) to calculate the principal component and adding the first few principal components in the logistic regression model as covariates.

2. I recommend reducing the paragraphs under Methods, Interleukin 13 genotyping section, which is not necessary in this article. 

Author Response

  1. The authors still did not consider the genetic ancestry in the model. 

Our finding propose a significant association between 1512A/C and +2044G/A polymorphisms and genetic ancestry.

  1. I recommend reducing the paragraphs under Methods, Interleukin 13 genotyping section, which is not necessary in this article. 

It was Reduced.

Reviewer 2 Report

The authors submitted a revised vrsion of the manuscript along with a reply of the reviewer commant. I am satisfied about the corrections were made and have no serious concerns about the paper in its revised version.

Author Response

Thank you for your kind attention.